# Toward Explainable AutoEncoder-Based Diagnosis of Dynamical Systems

**Gregory Provan**

School of Computer Science and IT, University College Cork (UCC), T12 R229 Cork, Ireland; g.provan@cs.ucc.ie; Tel.: +353-879549678

**Abstract:** Autoencoders have been used widely for diagnosing devices, for example, faults in rotating machinery. However, autoencoder-based approaches lack explainability for their results and can be hard to tune. In this article, we propose an explainable method for applying autoencoders for diagnosis, where we use a metric that maximizes the diagnostics accuracy. Since an autoencoder projects the input into a reduced subspace (the code), we define a theoretically well-understood approach, the subspace principal angle, to define a metric over the possible fault labels. We show how this approach can be used for both single-device diagnostics (e.g., faults in rotating machinery) and complex (multi-device) dynamical systems. We empirically validate the theoretical claims using multiple autoencoder architectures.

**Keywords:** diagnosis; autoencoder; subspace projection; principal angle

## 1. Introduction

### 1.1. Autoencoders in Diagnosis

Data-driven diagnosis using deep learning is starting to have a big impact in diagnostics. Applications have been developed for applications such as rotating machinery [1,2], as well as for other areas, such as medicine [3]. The common theme in most of these applications is the use of image data (e.g., X-rays in medicine) or time-series data. Hence, few applications that apply to complex systems, such as large process-control systems, have been developed.

One of the most common deep learning architectures for diagnosis is the autoencoder [2]. The autoencoder (AE) has been used for fault detection and for fault isolation. AE performance is comparable to using traditional signal-processing methods, but it has been supplanting these methods since data pre-processing is not needed in AE-based approaches, whereas it is a manual and time-consuming requirement for traditional signal-processing methods. A major drawback to AE-based approaches is the need for large volumes of (labeled) data, and the lack of explainability for the results.

Ideally, we use an AE as both a residual generator [4] and decision-logic tool to isolate faults. The question is how to train the AE to generate good diagnostic results, i.e., to best distinguish among the different failure conditions (or modes).

An AE has two steps: (1) *the encoding step* transforms the input $\xi \subset \mathcal{X}$ into a representation $\zeta = f(\xi)$ that resides in a subspace $\mathcal{W}$ of $\mathcal{X}$; (2) *the decoding step* performs the inverse transform, performing the mapping $\xi' = f(\zeta)$. The standard metric (or loss function) used in an AE is the $L_2$, or mean-squared error (MSE), metric defined over the input $\xi$ and reconstructed input $\hat{\xi}$, i.e., $(\xi - \hat{\xi})^2$. For diagnostics purposes, the goal is that the subspace $\mathcal{W}$ will enable us to classify different fault modes, i.e., it disentangles $\xi$ [5,6]. The key to successfully using an AE for diagnosis is to identify the subspace $\mathcal{W}$ that best distinguishes the different nominal and failure modes of the system under analysis. Unfortunately, this $L_2$ loss function does not have clear semantics for diagnostics isolation in comparison to a metric based on an encoded subspace whose principal angle [7] best distinguishes the

nominal and failure modes of the system. An example of such a metric is the gap metric [8]. In comparing a nominal input $\zeta$ and new input $\hat{\zeta}$, the gap metric is $[0, 1]$-bounded, and has well-defined interpretations for 0 (behaviors $\zeta$ and $\hat{\zeta}$ are identical) and 1 (behaviors $\zeta$ and $\hat{\zeta}$ are maximally different) [8]. In contrast, two behaviors can be similar even though their $L_2$ metric can be arbitrarily large ([9], p. 349).

### 1.2. Contributions

In this article, we use an autoencoder with a gap metric to provide clear semantics for diagnostics applications. The core idea that we exploit is the use of deep kernel architectures for the projection of fault-based information from data, and a representation of diagnosis metrics using principal angles. We show that the success of AE-based approaches to diagnosis depends on identifying an encoded subspace whose principal angle [7] best distinguishes the nominal and failure modes of the system. The principal angle between two vectors in a vector space $\mathcal{X}$ specifies a notion of canonical correlation between those vectors in a subspace $\mathcal{W} \subseteq \mathcal{X}$, and enables the generation of a distance metric between vectors. As a consequence, we can use this distance metric to distinguish vectors representing nominal and failure modes: vectors with similar $\delta$ will tend to be in the same mode, and with the metric that it is far away, it will be in different modes. This new metric thus provides more accurate diagnostics performance on the experiments conducted, compared to the traditional $L_2$ metric.

Our contributions are as follows.

1.  We define an explainable autoencoder in terms of a gap metrics, whose induced subspace generates a principal angle [7] to best distinguish the nominal and failure modes of the system.
2.  We show how an autoencoder can diagnose linear time-invariant (LTI) dynamical systems. We extend the application of AE-based approaches to time-series data and dynamical systems, and we provide a clear semantics (and hence explainability) for such approaches.
3.  We illustrate our approach using examples from a time-series benchmark, the Numenta Anomaly Benchmark (NAB) dataset [10,11].

This article is organized as follows. Section 2 compares and contrasts related work of relevance. Section 3 introduces the technical material for this article, overviewing dynamical systems, autoencoders and the temporal sequence that we use for diagnostics input. Section 4 defines the metrics that we adopt in this article. Section 5 describes our methodology that we use for computing AE-based diagnosis via metrics over subspaces. Section 6 presents our empirical analysis, where we compare standard and SPA metrics in terms of diagnostics accuracy.

## 2. Related Work

There is significant work related to this paper. We look at work on the theoretical basis we adopt, gap metrics, and on the various applications of AE-based methods.

### 2.1. Gap Metrics and AE-Based Diagnosis

Some recent papers have applied autoencoders and gap metrics for diagnostics. Ref. [12] shows how gap metric techniques can be applied to fault detection performance analysis and fault isolation schemes. Ref. [13] presents an integrated model-based and data-driven gap metric method for fault detection and isolation. Ref. [14] describes a related fault detection approach, which is based on a data-driven K-gap metric [12]. This has been applied to ship propulsion systems. Ref. [15] details a comparative study of K-gap metric-based techniques, based on different data-driven stable kernel representation methods. Ref. [16] empirically evaluates gap-metric based, multi-model control schemes for nonlinear systems.

### 2.2. Diagnosis Applications of AEs

AEs have been applied to a number of different diagnosis applications. The typical types of input data are time-series and image, although some categorical/table-based inputs have also been studied.

In typical diagnosis applications, an AE is augmented with a threshold to assess whether data are faulty or not. Ref. [17] extends this notion and proposes a latent reconstruction error metric as a *health indicator* for machine condition monitoring, which is built in the latent space of a deep autoencoder. Ref. [17] defines the latent reconstruction error of a sample $\xi_i$ as the Euclidean distance between its latent representation $\zeta_i$ and the latent representation of its reconstruction $\hat{\zeta}_i$: $d(\xi) = \|\zeta_i - \hat{\zeta}_i\|_2$.

#### 2.2.1. Diagnosis of Individual Machines

The most common time-series application domain is that of rotating machinery. Ref. [18] presents a survey on deep learning-based bearing fault diagnosis, in which AEs play a prominent role. Many papers have been published on applying autoencoders to rotating machinery fault diagnosis, such as [1,2,19,20].

We can contrast our proposed AE-based method with a closely related method [21], which uses a similar Hankel matrix-based input for the condition monitoring and diagnosis of rolling element bearings. In this approach, they compute anomalies using the eigenvalues of the input Hankel matrix and use as a metric the Frobenius norm.

In comparison to these other papers, our approach should have higher isolation accuracy than any approach that does not use a similar metric space.

#### 2.2.2. Data-Driven Diagnosis of Complex Dynamical Systems

A significant body of work has been generated on the data-driven analysis of LTI systems, based around the seminal work of [22]. This area is referred to as the behavioral approach to systems theory; see [23] for a survey and tutorial introduction. The behavioral approach considers a dynamical system not in state–space form but as a set of trajectories, based on a generator defined by LTI properties.

Recent work has applied subspace methods to the analysis of sets of trajectories, e.g., [23]; however, no AE-based solutions have been proposed yet within this behavioral framework.

### 2.3. Other Applications

*Image Inputs:* Ref. [24] surveys the use of deep autoencoders in pattern recognition. The most common application domain is that of medical diagnosis, e.g., X-ray or other medical imaging data. Ref. [25] surveys research on the application of autoencoder algorithms to diagnose rare diseases.

*Prognosis:* Autoencoders have recently found use in prognostics. Ref. [26] surveyed the use of variational autoencoders for prognosis and health management of industrial systems. Ref. [27] propose a prognostic health indicator, defined using Kullback–Leibler divergence, and applied to predict the remaining useful life of concrete structures.

## 3. Preliminaries

### 3.1. LTI System

We focus on discrete-time linear time-invariant (LTI) dynamical systems in this article. We can define an LTI system [28] in state space form as

$$
\begin{aligned}
x_{k+1} &= Ax_k + Bu_k + w_k; x(0) = x_0; \\
y_k &= Cx_k + Du_k,
\end{aligned}
$$

(1)
(2)

where $x \in \mathbb{R}^n$ is the state vector, $x_0$ is the initial condition of the system, $w_k \sim N(0, Q)$ is the uncorrelated zero-mean Gaussian measurement noise and standard deviation $Q$, and $u$ and $y$ are the plant input and output vectors, respectively. Matrices $A$, $B$, $C$, $D$ are real constant matrices of appropriate dimensions.

We can extend the nominal system (Equations (1) and (2)) to incorporate faults. Faults may be additive or multiplicative, and we consider faults to both actuators and sensors. If we define multiplicative fault parameter matrices $\Gamma_a$, $\Gamma_s$ to both actuators and sensors, respectively, we obtain

$$
\begin{aligned}
x_{k+1} &= Ax_k + B\Gamma_a u_k + w_k; x(0) = x_0; & (3) \\
y_k &= \Gamma_s(Cx_k + Du_k), & (4)
\end{aligned}
$$

The fault matrices are diagonal matrices with the $[0, 1]$ parameters along the diagonal denoting the fault extent, with 0 denoting totally faulty and 1 normal. For example, we may have actuator loss of effectiveness encoded by this approach. We may define additive faults in an analogous manner.

We refer to a *mode* as an operating condition of an LTI system, as characterized by the system parameters $(A, B, C, D, \Gamma_a, \Gamma_s)$.

### 3.2. AutoEncoders

This section introduces the autoencoder and reviews how it has been used for diagnosis. We first describe the vanilla (or traditional) AE, and the outline the standard variants, namely the denoising autoencoder (DAE), sparse AE, contractive autoencoder (CAE), and variational autoencoder (VAE). An introduction of AEs is presented in [29,30].

Autoencoders have widely been used for anomaly detection and fault detection [1]. Given data $\xi$, a vanilla autoencoder $\mathcal{A}$ consists of two parts: (1) an encoder $f_{\theta_E}(\xi) = h$ that creates the latent code $h$ in terms of parameters $\theta_E$; (2) a decoder $g_{\theta_D}(h) = \hat{\xi}$ that re-creates the input from the latent code $h$ in terms of decoder parameters $\theta_D$.

Figure 1 depicts an autoencoder for diagnosing faults in time-series systems. An autoencoder projects the input data $\xi$ into a smaller subspace, called the code $z$, such that linear classifiers can be used on the structure of the code. Autoencoders can thus perform nonlinear dimensionality reduction, as well as classification. Mathematically, learning an autoencoder can be expressed as the following optimization task:

$$
\min_{f,g} \parallel \xi - g \circ (f(\xi)) \parallel_2^2 \tag{5}
$$

When we train $\mathcal{A}$ with nominal data, then an input $\xi$ that is anomalous will produce a higher than average anomaly score. However, in many AE-based methods, the loss function $\mathcal{L}$ is typically defined in terms of MSE, or of a regularized balancing of (a) accuracy-based MSE and (b) some sparsity metric over the code $\mathcal{L} = \mathcal{L}_{MSE} + \lambda\mathcal{L}_{sparsity}$, where $\lambda$ is the regularization parameter.

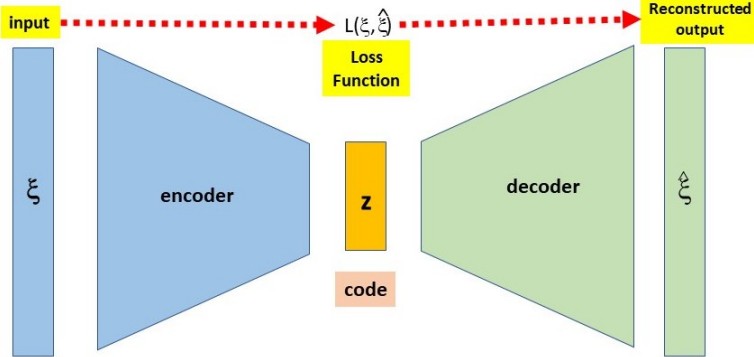

**Figure 1.** The autoencoder approach we adopt, where the input $\xi$ is a encoded to a code $z$ and then decoded to create the reconstructed input $\hat{\xi}$. We use the loss function $L(\xi, \hat{\xi})$ to control the quality of the reconstruction.

We can use $\mathcal{A}$ as a decision tool to identify faults, where an input $\xi$ is anomalous if some measure (an anomaly score) over the reconstruction $\hat{\xi}$ and input $\xi$ by more than a threshold $\epsilon$. Several definitions of the anomaly score have been proposed, such as reconstruction error ($\epsilon_R = |\xi - \hat{\xi}|$), or Mahalanobis distance, $\epsilon_M = \sqrt{(\hat{\xi} - \xi)^T \Sigma^{-1} (\hat{\xi} - \xi)}$, where $\Sigma$ is the covariance matrix of the training input.

Other AE approaches add regularization to *reduce and/or minimize the size* of the latent code. Common approaches include the sparse autoencoder (which minimizes the number of active neurons in the code, i.e., $\min_{z \in \zeta} |z|$), or the contractive autoencoder (which limits the sensitivity to small changes in the input, using the Frobenius norm $\| \cdot \|_F$ of the Jacobian matrix of the encoder).

A problem with using $\mathcal{A}$ is the lack of semantics for the approach, and the challenge of scaling to complex systems, e.g., dynamical systems such as process-control factories. Further, the loss function associated with Equation (5) best addresses single-fault diagnosis (fault/no-fault) cases: it aims to match the input $x$ with the output $\hat{x}$. This representation focuses on identifying an anomalous sequence; hence, it does not address fault isolation.

### 3.3. Temporal Sequence Representation

As LTI systems are universal approximators of temporal sequences [31], a temporal sequence can be regarded as the output of an LTI system of unknown parameters. Hence, we can use a temporal sequence to diagnose an LTI system.

Assume that we have a system $\mathcal{S}$ that generates a language $\mathbb{B}$ consisting of valid temporal sequences, or behaviors. A behavior is a temporally indexed sequence $\xi = \{\xi_1, \ldots, \xi_T\}$; a behavior is valid if $\xi$ is consistent with an underlying system model, e.g., as described by Equations (1) and (2). In the following, we consider that each behavior is valid, i.e., $\xi \in \mathbb{B}$. In this article, we assume that each behavior is time invariant, linear, and has bounded zero-mean noise. We define a vector $\xi|_\lambda$ as a $\lambda$-length trajectory of a signal.

We can represent a signal of length $\lambda \in T$ in terms of a data matrix. We adopt a Hankel matrix of depth $\lambda \in T$ for signal $\xi \in \mathbb{R}^{qT}$, as follows:

**Definition 1** (Hankel matrix).

$$H_\lambda(\xi) = \begin{bmatrix} \xi_1 & \xi_2 & \cdots & \xi_{T-\lambda+1} \\ \xi_2 & \xi_3 & \cdots & \xi_{T-\lambda+2} \\ \cdots & \cdots & \cdots & \cdots \\ \xi_\lambda & \xi_{\lambda+1} & \cdots & \xi_T \end{bmatrix} \tag{6}$$

*where $\xi = [\xi_1, \xi_2, \ldots, \xi_T]$ denotes an acquired signal sequence, $\lambda$ is the embedding dimension, and $T - \lambda + 1$ is the length of each sub-sequence.*

With this input encoding, the size of the associated Hankel matrix $H$ is $q\lambda \times (T - \lambda + 1)$. As can be seen in Definition 6, if the Hankel matrix $H$ is square, $H$ is also a symmetric matrix. The Hankel matrix, as can be seen in Definition 6, encodes a time-series signal with minimal pre-processing, and hence places limited computational burden on the overall inference process.

The Hankel matrix $H$ embeds the observability matrix $\Gamma$ of the system, since $H = \Gamma X$, where $X$ is the sequence of hidden states of the LTI system, and $\Gamma$ is defined as follows:

$$\Gamma = \begin{bmatrix} C \\ CA \\ \cdots \\ CA^\lambda \end{bmatrix} \tag{7}$$

Therefore, $H$ provides information about the dynamics of the temporal sequence. Given an input–output measurement $(u, y)$, we can estimate $A, C$, and the initial state $x_0$ to

identify the corresponding LTI system for classification purposes. The identification of the triple is, however, a non-convex problem and thus, given a finite measurement $(u, y)_T$, the triple $(A, C, x_0)$ used to generate such a measurement is not guaranteed to be unique [32]. Consequently, system identification is computationally expensive and not robust, which makes it unsuitable for diagnosis purposes.

These problems can be avoided by using subspace identification on Hankel matrices associated with the measurement signals [33]. Here, given a dynamical system, all output measurements lie on a single subspace, assuming a noiseless output. This means that the subspace spanned by the columns of a Hankel matrix is equivalent to the subspace of the associated LTI system. Therefore, the subspace spanning an LTI system can be computed and used for diagnosis, without identifying the underlying LTI system.

Equation (7) shows that the columns of $H$ and $\Gamma$ span the same subspace regardless of the initial values of the LTI system. This means that given two measurements $\xi$ and $\xi'$ from the same LTI system operating in the same mode, the smallest principal angle between the subspaces of the Hankel matrices $H(\xi)$ and $H(\xi')$ is zero [32]. In other words, the subspace angles between the Hankel matrices of two output measures can be used to identify whether these outputs could be produced by the same LTI system. Even though the relation between output signals and LTI systems is non-unique, for diagnosis purposes, we assume that two Hankel matrices which share the same subspace belong to the same LTI system, and thus belong to the same diagnosis mode (or class label in learning terms).

## 4. Gap Distance Metric

### 4.1. Diagnosis via Sub-Space Analysis

This section describes how we compute diagnoses via sub-space projections. We first introduce our diagnosis task over a Hilbert space. A Hilbert space $\mathcal{H}$ is a vector space with an inner product $\langle \cdot, \cdot \rangle$. Given a vector $\mu \in \mathcal{H}$, we compute a metric using the norm of $\mu$, which is induced by the inner product, $\| \mu \| = \mu \cdot \mu$.

**Definition 2** (Diagnosis Task). *We define a classification (e.g., diagnosis) problem in Hilbert space as follows: given a subspace $\mathcal{J} \in \mathcal{H}$, check if $\mu \in \mathcal{H}$ belongs to $\mathcal{J}$.*

We solve this task using two steps: (i) project $\mu$ onto $\mathcal{J}$; and then (ii) calculate the distance between $\mu$ and its projection using the induced norm.

This notion of inner products is generalized through the notion of kernels. Kernel methods [34,35] have been used extensively in machine learning for classification, i.e., to capture the nonlinear complex patterns underlying data. Kernel methods include support vector machines (SVMs), kernel Fisher discriminant analysis (KFDA), and Gaussian process models. They have been successfully applied to a wide variety of machine learning problems [34,35]. These methods map data points from the input space to the feature space, i.e., higher dimensional reproducing kernel Hilbert space (RKHS), such that relatively simple linear algorithms can deliver impressive performance.

**Definition 3** (Kernels via feature maps). *Let $\Omega$ be a nonempty set. A feature map $\phi$ is any function $\phi : \Omega \to \mathcal{H}$, where $(\mathcal{H}, \langle \cdot, \cdot \rangle_{\mathcal{H}})$ is any Hilbert space (the feature space). The function*

$$\mathcal{K}(x, y) := \langle \phi(x), \phi(y) \rangle_{\mathcal{H}} \quad x, y \in \Omega,$$

*is a positive definite kernel on $\Omega$.*

Denoting the $d$-dimensional input space $\mathcal{X} \in \mathbb{R}^d$ ($\mathbb{R}$ denotes the set of real numbers), the kernel function $\mathcal{K} : \mathcal{X} \times \mathcal{X} \to \mathbb{R}$ induces an implicit mapping $\phi$ into a higher dimensional RKHS $\mathcal{H}$ in which even simple linear models inferred are highly effective compared to their nonlinear counterparts learned directly in the input space $\mathcal{X}$.

Instead of formulating an optimization criterion with a fixed kernel $\mathcal{K}$, one can leave the kernel $\mathcal{K}$ as a combination of a set of predefined kernels, which results in the problem of

multiple kernel learning (MKL) [36]. MKL maps each sample to a multiple-kernel-induced feature space, and a linear classifier is learned in this space.

So far, we do not know which kernel will produce the most discriminative AE in terms of diagnostics isolation. We now show how the notion of principal angle provides a metric for the necessary kernel function.

*4.2. Metrics for Temporal Sequences*

We can extend this notion to temporal sequences of vectors as follows. Assume that we have a system $\mathcal{S}$ that generates a language $\mathbb{B}$ consisting of valid behaviors. We define length-$\lambda$ behaviors as subspaces of equal dimension, which may be represented directly by data matrices. Thus, length-$\lambda$ behaviors may be identified with points on the Grassmannian $Gr(k, N)$, i.e., the set of all subspaces of dimension $k$ in $\mathbb{R}^\lambda$, endowed with the structure of a (quotient) manifold.

**Proposition 1.** *The function d is a metric on the set of all restricted behaviors $\xi|_\lambda$, with $\lambda > l$, whenever d is a metric on $Gr(m\lambda + n, q\lambda)$.*

**Definition 4** (Gap Function Projection Operator *P*). *Let $\mathcal{H} = \mathcal{X} \times \mathcal{Z}$, with $\mathcal{X}$ and $\mathcal{Z}$ Hilbert spaces. Let $P : dom(P) \to \mathcal{Z}$ and $\tilde{P} : dom(\tilde{P}) \to \mathcal{Z}$ be closed operators, with $dom(P)$ and $dom(\tilde{P})$ being subspaces of $\mathcal{X}$. The gap between P and $\tilde{P}$ is defined as*

$$gap_{\mathcal{H}}(P, \tilde{P}) = gap_{\mathcal{H}}(\mathcal{G}(P), \mathcal{G}(\tilde{P})). \tag{8}$$

**Definition 5.** *The set of all behaviors $\xi|_\lambda$, with $\lambda > l$, equipped with gap $\lambda$ is a metric space.*

The underlying Grassmannian structure enables us to induce metrics between behaviors. We represent the distance between behaviors $\xi$ and $\tilde{\xi}$ using the function $gap_\lambda(\xi, \tilde{\xi})$.

Dynamical systems projections are viewed as a graph, which is defined as follows:

**Definition 6** (Graph). *Let $T : \mathcal{X} \to \mathcal{Z}$ be a map between sets. Then the graph $\mathcal{G}(T) \subseteq \mathcal{X} \times \mathcal{Z}$ is given by $\mathcal{G}(T) = \{(\xi, T\xi) : \xi \in \mathcal{X}\}$. If T is linear then $\mathcal{G}(T)$ is a linear subspace of $\mathcal{X} \times \mathcal{Z}$.*

The gap metric has a well-known geometric interpretation in terms of the sine of the largest principal angle between two subspaces. We define the principal angle as follows.

**Definition 7** (Principal Angle). *Consider two subspaces $\mathcal{X}$ and $\mathcal{Z}$ of $\mathbb{R}^n$ with dimensions r and s respectively, where $r \leq s$. The principal angles between $\mathcal{X}$ and $\mathcal{Z}$, denoted as $\theta_1, \dots, \theta_r$, are defined recursively as follows for $j = 2, \dots, r$.:*

$$\theta_1 = \min_{\mathbf{x}_1 \in \mathcal{X}, \mathbf{z}_1 \in \mathcal{Z}} \arccos\left( \frac{\mathbf{x}_1^\top \mathbf{z}_1}{\|\mathbf{x}_1\| \|\mathbf{z}_1\|} \right),$$

$$\vdots$$

$$\theta_j = \min_{\substack{\mathbf{x}_j \in \mathcal{X}, \mathbf{z}_j \in \mathcal{Z} \\ \mathbf{x}_j \perp \mathbf{x}_1, \dots, \mathbf{x}_{j-1} \\ \mathbf{z}_j \perp \mathbf{z}_1, \dots, \mathbf{z}_{j-1}}} \arccos\left( \frac{\mathbf{x}_j^\top \mathbf{z}_j}{\|\mathbf{x}_j\| \|\mathbf{z}_j\|} \right)$$

The vectors $x_1, \dots x_r$ and $z_1, \dots z_r$, are called principal vectors. The dimension of $\mathcal{X} \cap \mathcal{Z}$ is the multiplicity of zero as a principal angle. It is straightforward to compute the principal angles by calculating the singular values of $X^T Z$, where $X$ and $Z$ are orthonormal bases for $\mathcal{X}$ and $\mathcal{Z}$, respectively. The singular values of $X^T Z$ are then $cos\theta_1, \dots, cos\theta_r$.

A principal angle induces several distance metrics on the Grassmann manifold. One example is the (squared) chordal distance $\mathscr{D}_c^2(\mathcal{X}, \mathcal{Z})$, given by

$$\mathscr{D}_c^2(\mathcal{X}, \mathcal{Z}) = \sum_{i=1}^s \sin^2 \theta_i.$$

Hence, when $\lambda > l$, the $\lambda$-gap between $\xi|_\lambda$ and $\tilde{\xi}|_\lambda$ is $gap_\lambda(\xi, \tilde{\xi}) = sin\theta_{max}$, where $\theta_{max}$ is the largest principal angle between the sub-spaces $\xi|_\lambda$ and $\tilde{\xi}|_\lambda$.

One key property of such a metric is that it generates admissible systems of neighborhoods; this enables the classification of different modes by neighborhood. We shall call such metrics robust [8].

There are many variants of the gap metric in the literature, e.g., [37–39]. All of these metrics are equivalent, and induce the same graph topology.

**Theorem 1** ([8], Theorem 7). *Every robust metric creates the same topology as the gap metric.*

The induced metrics differ in terms of advantages and disadvantages in particular applications and analyses. The standard autoencoder uses an $L_2$ (norm) metric; this metric is less explainable than than the gap metric since the gap metric is $[0, 1]$-bounded with clear interpretations for 0 (behaviors $\xi$ and $\tilde{\xi}$ are identical) and 1 (behaviors $\xi$ and $\tilde{\xi}$ are maximally different) [8]. In contrast, two behaviors can be similar even though the $L_2$ metric between them can be arbitrarily large ([9], p. 349).

We will focus on a principal angle metric in the following.

*4.3. SVD of Hankel Matrix*

Once we encode a signal of length $\lambda \in T$ in terms of a Hankel matrix of depth $\lambda \in T$ for signal $x \in \mathbb{R}^{qT}$, the size of the associated Hankel matrix $\boldsymbol{H}$ is $qL \times (T - \lambda + 1)$. As can be seen in Definition 6, if the Hankel matrix $\boldsymbol{H}$ is square, $\boldsymbol{H}$ is also a symmetric matrix.

The $\lambda$-gap between behaviors can be directly computed from the knowledge of trajectories. Let $\xi$ and $\tilde{\xi}$ be T-length trajectories of order $\lambda$, with $\lambda > l$. Let

$$\boldsymbol{H}_\lambda(\xi) = \begin{bmatrix} U_1 & U_2 \end{bmatrix} \begin{bmatrix} S & 0 \\ 0 & 0 \end{bmatrix} \begin{bmatrix} V_1 \\ V_2 \end{bmatrix}$$

$$\boldsymbol{H}_\lambda(\tilde{\xi}) = \begin{bmatrix} \tilde{U}_1 & \tilde{U}_2 \end{bmatrix} \begin{bmatrix} \tilde{S} & 0 \\ 0 & 0 \end{bmatrix} \begin{bmatrix} \tilde{V}_1 \\ \tilde{V}_2 \end{bmatrix}$$

be the singular value decomposition (SVD) of the Hankel matrices $\boldsymbol{H}_\lambda(\xi)$ and $\boldsymbol{H}_\lambda(\tilde{\xi})$ with $U_1 \in \mathbb{R}^{q\lambda \times (m\lambda+n)}$ and $\tilde{U}_1 \in \mathbb{R}^{q\lambda \times (m\lambda+n)}$, respectively. Then

$$gap_\lambda(\xi, \tilde{\xi}) = \|U_1 U_1^T - \tilde{U}_1 \tilde{U}_1^T\|_2 = \| \tilde{U}_2 U_1^T \|_2 \tag{9}$$

where the first equality comes from the fact that $gap_\lambda(\xi, \tilde{\xi}) = \| P_{\xi|_\lambda} - P_{\tilde{\xi}|_\lambda} \|_2$ and we substitute $P_{\xi|_\lambda} = U_1 U_1^T$ and $P_{\tilde{\xi}|_\lambda} = \tilde{U}_1 \tilde{U}_1^T$ [40].

**5. Approach**

This section describes our approach: we outline our input data, and how we transform those data for diagnostic purposes.

We assume that we measure data $\xi$, where each $\xi$ is a $q$-vector in a space $\mathcal{X} \subseteq \mathbb{R}^q$. A discrete-step time-series signal is denoted by a time set $Z^+$ and signal $\xi \subseteq \mathbb{R}^{Z+}$.

We assume that the data-generator (LTI system, such as a pump or motor) generates data with multiple different distributions; we call each of these a distinguished *mode*. In diagnostic terms, the modes correspond to nominal or faulty conditions of the motor. We define the set of modes as $\Gamma = \{\gamma_1, \ldots, \gamma_J\}$.

Given a data set $\mathcal{D}$, we want to train an autoencoder (AE) $\mathcal{A}_\theta$ with parameters $\theta$ in parameter set $\Theta$ to distinguish the data according to their mode distribution. Assume that we have a loss function that scores correctly diagnosing faults in inputs $\xi$ in a test set $\mathcal{D}_{test}$.

**Task 1** (Autoencoder Learning). *Our learning task is to generate the autoencoder parameters $\theta^*$ that minimize diagnostics loss over a test set while maximizing mode distance, given true fault label $\hat{y}_i$ for test instance i.*

$$\theta^* = \arg\min_{\theta \in \Theta} \max_{\gamma_j \in \Gamma} \sum_{i \in \mathcal{D}_{test}} \mathcal{L}(\hat{y}_i, A_\theta(\xi_i)), \tag{10}$$

*where $\mathcal{L}(\hat{y}_i, A_\theta(\xi_i))$ measures the loss between the true fault label $\hat{y}_i$ and the autoencoder output $A_\theta(\xi_i)$.*

We outline our approach in Figure 1. We train an AE and use a loss function $\mathcal{L}$ that aims to optimize diagnostic accuracy. Specifically, we encode a loss function $\mathcal{L}_{SPA}$ that generates a code based on the subspace principal angle (SPA). Here, we encode the input in a specific form (Hankel matrix), and perform AE-based diagnosis by defining the code as a representation that aims to maximize the distance of the SPA, $\zeta = f_\rho(h)$, as follows:

1. Encode the input data $\xi$ as a Hankel matrix.
2. Compute, from the SVD of the corresponding Hankel matrix $h = f_{SVD}(\xi)$, the principal angles $\rho$.
3. Compute subspace distance in terms of SPA.
4. Output the loss as the minimum SPA.

Previous work, e.g., [12,41,42], addressed system identification of an arbitrary LTI dynamical system $\mathcal{S}$, based on data consisting of input/output pairs $(u, y)$ such that $\mathcal{S}$ is controllable. In this case, we require that (a) our language $\mathbb{B}$ properly encodes the input/output relations of an LTI system $\mathcal{S}$, and (b) our inputs $u$ are persistently exciting (a signal is persistently exciting if its spectrum contains a sufficiently large number of harmonics [43]; the use of persistently exciting signals ensures that a system identification experiment produces informative data). In this article, we restrict the data to just outputs $y$.

## 6. Empirical Analysis

This section describes our empirical analysis of the impact of principal angle methods.

### 6.1. Experimental Design

We train the autoencoder using an optimization framework as follows. The primary objective is to minimize the difference between the input matrix $H$ and the reconstructed matrix $\hat{H}$. We then apply regularization to enforce the code $\zeta$, maximizing the metric distance between different modes. If we define $\xi^i$ and $\xi^j$ as behaviors from modes indexed by $i, j \in \mathcal{I}$, respectively, we can define this constraint as

$$\max_{i \neq j, \, i,j \in \mathcal{I}} \sum_{k=1}^{N} gap_\lambda(\xi^i, \tilde{\xi}^j) \tag{11}$$

We can also aim to restrict the size of the code $\zeta$, as is typically done in AE training.

We designed experiments to study how the AE architecture and loss function affect diagnostic performance. Table 1 summarizes the experiments that we conducted. We developed two main types of architecture, based on fully connected and long short-term memory (LSTM) architectures. We also compare the impact of different loss functions.

**Table 1.** Experiments to compare model types and loss functions. MAE stands for *mean absolute error* and PA for *principal angle*.

| Model | Loss Function | Remarks |
|---|---|---|
| fully connected | MAE | deep network with 2 hidden dense |
| fully connected | PA | layers in encoder/decoder |
| LSTM | MAE | deep network with 2 LSTM |
| LSTM | PA | layers per encoder/decoder |

We use the architecture to encode the time-series model. A fully connected architecture is the simplest approach, providing a clear method for the model, although it has not a specific structure for encoding temporal representations.

The LSTM is designed to encode temporal features, and hence should be better than a fully connected architecture for time-series models.

### 6.2. Data

We use the Numenta Anomaly Benchmark (NAB) dataset [10,11] for evaluating the prediction performance of the proposed framework. It provides artificial and real-world time-series data containing labeled anomalous periods of behavior. Data are ordered and timestamped, with single-valued metrics. The timestamp is of the format "yyyy-mm-dd hh.mm.ss" representing the year, month, day, hour, minute, and second. The difference of adjacent timestamps is fixed; however, such a difference varies for various datasets. An entry is labeled as anomalous if its timestamp is specified explicitly in a separate file.

Our AE-based approach conducts semi-supervised anomaly detection, so it does not rely on labels to train anomaly detection models; instead, we use the labels for performance evaluation. We use the art_daily_small_noise.csv file for training and the art_daily_jumpsup.csv file for testing. The data consist of 288 timesteps/day, i.e., a value every 5 min for 14 days. We use a batch size of 3745 sequences.

*Pre-processing*: We normalize the data prior to inference.

### 6.3. Architecture

In this article, we compare two different deep network architectures, which are summarized in Figures 2 and 3: a fully connected (dense) and an LSTM architecture. The dense network contains 10 times more trainable parameters than the LSTM network.

```
Model: ''model_2''
_________________________________________________________________
Layer (type)                 Output Shape              Param #
=================================================================
input_2 (InputLayer)         [(None, 288, 1)]          0

dense_6 (Dense)              (None, 288, 128)          256

dense_7 (Dense)              (None, 288, 64)           8256

dense_8 (Dense)              (None, 288, 32)           2080

dense_9 (Dense)              (None, 288, 64)           2112

dense_10 (Dense)            (None, 288, 128)          8320

dense_11 (Dense)            (None, 288, 1)            129

=================================================================
Total params: 21,153
Trainable params: 21,153
Non-trainable params: 0
```

**Figure 2.** Specification for AE with dense architecture.

```
Model: ''model_1''

_________________________________________________________________
Layer (type)                    Output Shape            Param #
=================================================================
input_3 (InputLayer)            [(None, 288, 1)]        0

lstm_4 (LSTM)                   (None, 288, 16)         1152

lstm_5 (LSTM)                   (None, 4)               336

repeat_vector_1 (RepeatVect     (None, 288, 4)          0
or)

lstm_6 (LSTM)                   (None, 288, 4)          144

lstm_7 (LSTM)                   (None, 288, 16)         1344

time_distributed_1 (TimeDis     (None, 288, 1)          17
tributed)

=================================================================
Total params: 2993
Trainable params: 2993
Non-trainable params: 0
```

**Figure 3.** Specification for AE with LSTM-based architecture.

### 6.4. Loss Function

As a baseline, we study the "traditional" AE loss function, MAE. This loss function tries to get the input $\xi$ and output $\hat{\xi}$ to agree, by minimizing $L = (\xi - \hat{\xi})^2$.

We encode the subspace approach in terms of a PA-based loss function.

- Input $\xi$: a $T$-length time-series vector of data.
- Transform $\xi$ into a Hankel matrix $\boldsymbol{H}$.
- Apply an SVD projection of $\boldsymbol{H}$ ([44] describes a kernel SVD algorithm for such a task).
- Compute the SPA metric $P_{SPA}(z)$ to maximize distance between the modes in this subspace.

### 6.5. Results

We now summarize our results. We rank the performance of the classification models using mean classification accuracy:

$$Accuracy = \frac{\#\text{Correctly classified observations}}{\#\text{Total Observations}} \times 100. \tag{12}$$

We average the results over 50 different experiments, as shown in Table 2. This table shows that the LSTM architecture outperforms the fully connected architecture. Second, the SPA metric outperforms the MAE metric for both architectures.

**Table 2.** Summary of performance of model types and loss functions. MAE stands for *mean absolute error* and SPA for *smallest principal angle*.

| Model | Loss Function | Accuracy |
|---|---|---|
| fully connected | MAE | 73 |
| fully connected | SPA | 79 |
| LSTM | MAE | 86 |
| LSTM | SPA | 92 |

Figures 4 and 5 show some details of the results for fully connected and LSTM-based AE architectures, respectively.

**Fully-connected AE architecture:** Figure 4a plots the true data against the predicted values, and shows that the model trained using MAE has significant discrepancies. In contrast, Figure 4b shows that the model trained using SPA has fewer discrepancies. In terms of test loss, Figure 4c shows that the model trained using MAE is worse than the model trained using SPA; for example, the MAE loss spans 0.5 through 1.5, whereas the loss for the SPA model spans 0.4–0.9.

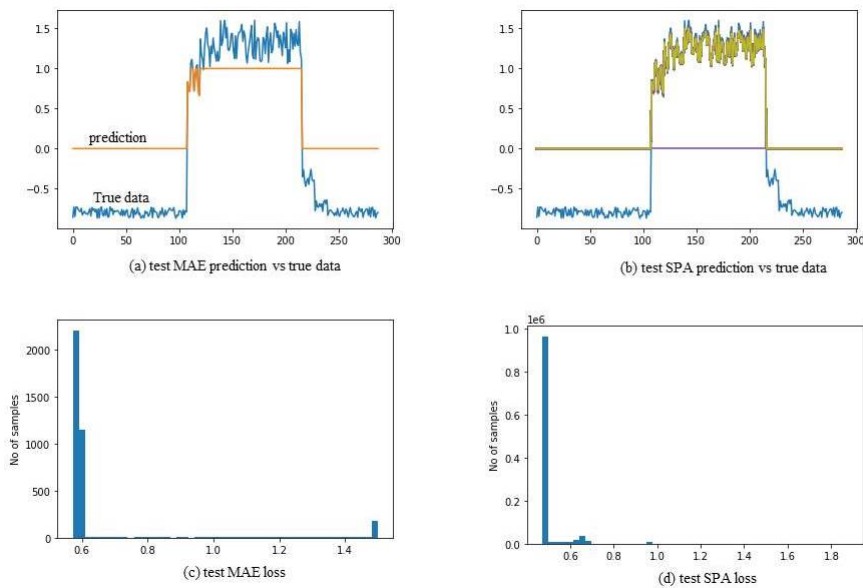

**Figure 4.** Results for the fully connected AE. Plots (**a**,**b**) show the true vs. predicted values for MAE and SPA loss functions, respectively. Plots (**c**,**d**) show the test loss for MAE and SPA loss functions, respectively.

**LSTM-based AE architecture:** Figure 5 shows that the LSTM-based architecture performs much better than the fully connected AE. Figure 5a plots the true data against the predicted values and shows that the model trained using MAE has greater discrepancies than that shown in Figure 5b, which shows the true vs. the predicted values for the model trained using SPA. In terms of test loss, Figure 5c shows that the model trained using MAE is worse than the model trained using SPA; for example, the MAE loss spans 0.13 through 0.6, whereas the loss for the SPA model (Figure 5d) spans 0.08–0.33.

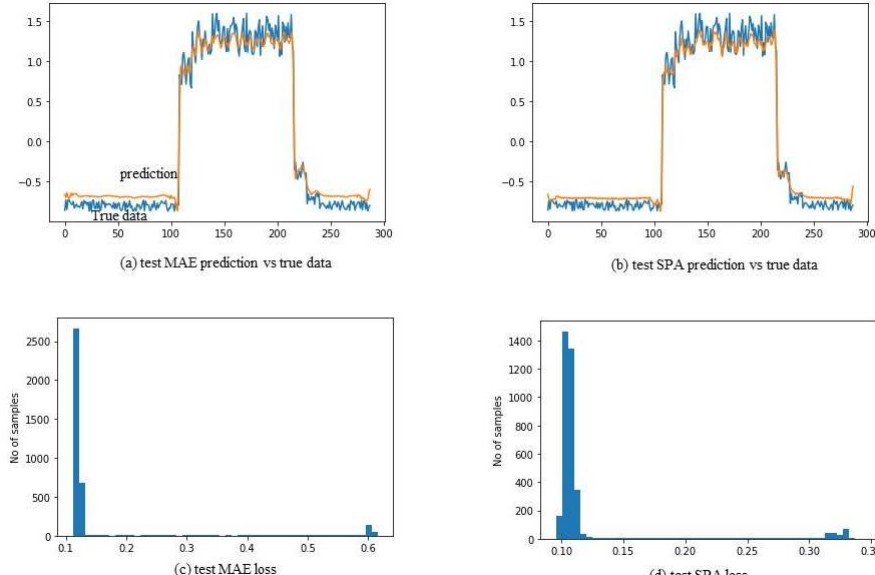

**Figure 5.** Results for the LSTM-based AE. Plots (**a**,**b**) show the true vs. predicted values for MAE and SPA loss functions, respectively. Plots (**c**,**d**) show the test loss for MAE and SPA loss functions, respectively.

## 7. Conclusions

We showed the theoretical underpinnings for a semantically well-defined AE for diagnostics inference on time-series data. The key element is the introduction of a well-defined metric space over the AE, based on the SPA. Traditional AE architectures are not trained with such a metric space, and hence can be shown to be sub-optimal with regard to diagnostics isolation accuracy.

We showed how the data-driven AE with SPA metrics can be applied to time-series isolation of machinery faults and of LTI systems. We empirically validated this theoretical position with experiments over two AE architectures.

This work provides the basis for improving the performance of AEs in diagnostics applications. Future work aims to further empirically justify this approach with additional real-world data sets.

**Funding:** This publication has emanated from research conducted with the financial support of Science Foundation Ireland (SFI) under Grant Number SFI/12/RC/2289 and 13/RC/2094.

**Data Availability Statement:** All data used in this study are publicly available at https://numenta.com/machine-intelligence-technology/numenta-anomaly-benchmark/ (accessed on 15 September 2022).

**Conflicts of Interest:** The authors declare no conflict of interest.

## Abbreviations

The following abbreviations are used in this manuscript:

| Abbreviation | Meaning |
|---|---|
| AE | Autoencoder |
| SVD | Singular Value Decomposition |
| SPA | Smallest Principal Angle |
| LTI | Linear Time Invariant |
| LSTM | Long Short-Term Memory |

| Symbol | Meaning |
| --- | --- |
| $x$ | state variable |
| $y$ | output variable |
| $u$ | input variable |
| $\zeta$ | time-series vector |
| $\boldsymbol{H}$ | Hankel matrix |
| $\theta$ | subspace angle |

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
