# Peer review of "Toward Explainable AutoEncoder-Based Diagnosis of Dynamical Systems"

_algorithms, doi:10.3390/a16040178_

Round 1

Reviewer 1 Report

Comments and remarks are in the attached file.
Yours sincerely

Author Response

I have made a significant rewrite of the article, as discussed in the attached file. Please let me know of the  opinions of the reviewers.

Kind regards,
Gregory Provan

===============================
Revisions made to Manuscript
1. The manuscript has been significantly rewritten, as per the reviews. We have reorganized the sections of the document to enhance legibility.
2. Notation: We have introduced tables of key abbreviations and symbols. We have made the notation throughout the document more consistent.
3. Section 2 (Related Work): this section has been extended and typos have been corrected.
4. Section 3 (Preliminaries): We carefully introduce all notation and bring together all the technical background necessary to understand the article.
5. Section 4 (Gap Distance): this section describes the key technical modifications to autoencoders, and has been extended to clarify the contributions.
6. Sections 5, 6 (Empirical Analysis): we have extended our analysis and our discussion of the results.

Reviewer 2 Report

This manuscript proposes an interpretable approach for applying autoencoders to fault diagnosis and using a metric to maximize diagnostic accuracy. The proposed approach is good but some important issues should be explained, to improve it further.

(1) It is important to have a thorough review of introduction, and I found that the authors did not give enough attention to the description of introduction.

(2) The resolution of the figure 1 is not clear, the resolution of the figure format should not be lower than 300*300dpi.

(3) Please list in detail the sources of the parameters required in the table. Numerical simulation data must be accurate.

(4) In general, from this empirical analysis, I find the validation of this approach unconvincing.

(5) The conclusion needs to be significantly revised to indicate your contribution to the interpretability of the autoencoder approach.

(6) The whole manuscript needs further polishing to correct grammar errors and typos.

Author Response

(The authors gave the same response as above.)

Round 2

Reviewer 1 Report

The paper is available for publication.

Reviewer 2 Report

This paper can be accepted in its current form.